# Coastal Retreat Due to Thermodenudation on the Yugorsky Peninsula, Russia during the Last Decade, Update since 2001–2010

**Marina Leibman** [1],*[ID], **Alexander Kizyakov** [2][ID], **Yekaterina Zhdanova** [3], **Anton Sonyushkin** [4] and **Mikhail Zimin** [5][ID]

1   Laboratory of Cryogenic Processes, Earth Cryosphere Institute, Tyumen Scientific Centre SB RAS, Malygin Street, 86, P.O. Box 1230, 625000 Tyumen, Russia
2   Cryolithology and Glaciology Department, Faculty of Geography, Lomonosov Moscow State University, GSP-1, Leninskie Gory, 119991 Moscow, Russia; akizyakov@mail.ru
3   Meteorology and Climatology Department, Faculty of Geography, Lomonosov Moscow State University, 119991 Moscow, Russia; ekaterinazhdanova214@gmail.com
4   OpenDataBox, 115573 Moscow, Russia; asonyushkin@opendatabox.info
5   Laboratory of Aerospace Methods, Department of Cartography and Geoinformatics, Faculty of Geography, Lomonosov Moscow State University, 119991 Moscow, Russia; ziminmv@mail.ru
*   Correspondence: moleibman@gmail.com; Tel.: +7-495-939-3673

**Abstract:** Thermodenudation on the Kara seacoast, the Yugorsky Peninsula, Russia, is studied by analyzing remote-sensing data. Landforms resulting from the thaw of tabular ground ice, referred to as thermocirques, are formed due to polycyclic retrogressive thaw slumps, during the last decade 2010–2020. We calculate the retreat rate of the thermocirque edge using various statistical approaches. We compared thermocirque outlines by the end of each time interval defined by the dates of available very-high-resolution imagery. Six thermocirques within two key sites on the Yugorsky peninsula are monitored. We correlate each of the thermocirque edge's retreat rates to various climatic parameters obtained at the Amderma weather station to understand the interrelation patterns better. As a result, we find a very low correlation between the retreat rate of each thermocirque and summer warmth, rainfall, and wave action. In general, the activity of thermodenudation decreases in time from the previous decade (2001–2010) to 2010–2020, and from 2010 towards 2020, although the summer warmth trend increases dramatically. A single thermocirque or series of thermocirques expand in response to environmental and geological factors in coastal retreat caused by thermodenudation.

**Keywords:** photogrammetric processing; multi-temporal remote sensing data; permafrost processes; retrogressive thaw slump; tabular ground ice; thermocirque; thermodenudation; Kara seacoast

## 1. Introduction

Slope failures in the continuous permafrost zone are tightly linked to ground ice distribution, especially to the melting of massive ground ice bodies [1–7]. In the last decade, several climate extremes were noted in the Arctic [8]. They attracted interest to the response of permafrost with massive ground ice to high summer warmth episodes with both positive (higher retreat rates) [9–15] and negative (lower retreat rates) [16] responses.

Additional factors that affect coastal retreat are wave action through coastal thermo-erosion and heavy rains that remove landslide bodies from the beach [4,5,17,18]. Snow accumulation accelerates nivation and thermoerosion [2,10] and controls ground temperature rise [19,20].

Coastal dynamics results from the activation of thermodenudation, the most widespread process in the area with tabular ground ice enclosed in geological sequences. The importance of the massive (tabular) ground ice occurrence as a major factor in coastal retreat is recognized in the literature [7,21–25].

We marked thermodenudation as the dominating coastal destruction mechanism in the study area [24]. Taken the process's polycyclic pattern, we use different terms for the landslide landform and landsliding as a process instead of one term retrogressive thaw slump (RTS). Term RTS causes complications when studying various stages and mechanisms of coastal retreat and slope mass waste. In Russian literature, we use the term thermodenudation, meaning tabular ground ice thaw and slope mass waste for the process of RTS formation. We use the term earth flow or RTS for a single feature produced by this process. We use the term "thermocirque" (TC) for an extensive landform resulting from a series of multiple-aged RTS occupying various positions within one landform (Figure 1).

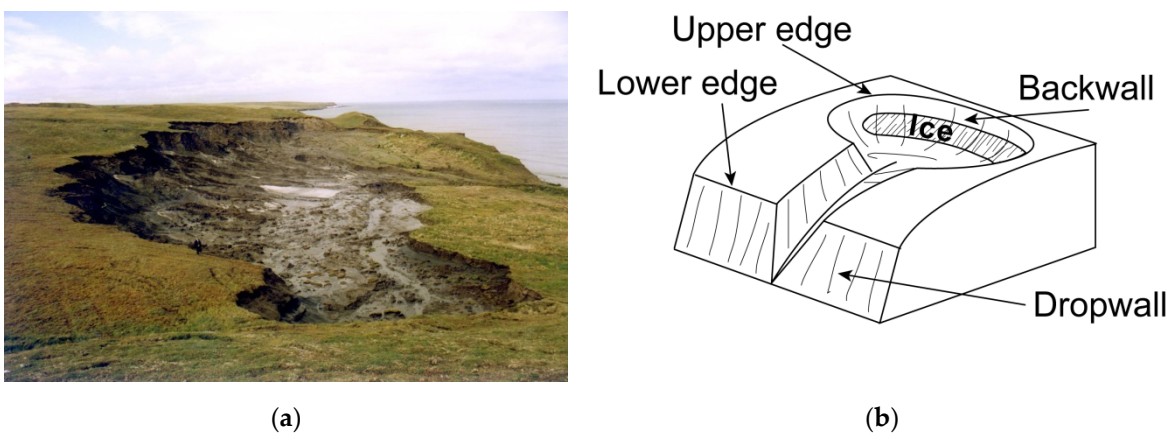

(**a**)           (**b**)

**Figure 1.** Thermocirques: (**a**) Western thermocirque appearance at the Pervaya Peschanaya key site; (**b**) thermocirque elements.

In this paper, we will use the term RTS only for the single earth flow path. TC at the coasts are step-shaped with two retreating planes and respective edges: a TC backwall with retreating "upper" edge, equal to headwall+headscarp after [13], and a dropwall to the beach with retreating "lower" edge (Figure 1b). Monitoring by remote-sensing data analysis allows most reliably locating the upper edge of TC and measuring its retreat rate.

The Yugorsky peninsula coast of the Kara Sea near Amderma settlement was studied since 1998. It was established then that warmer summers resulted in an increased rate of TC growth [2,10,24,26–32]. Our particular interest in the study of coastal retreat due to thermodenudation at Yugorsky coast is related to a probable reduction in the retreat in the last decade due to warming that could have caused insulation of ice exposures by excessive amounts of thawed material, specifically for low headwalls as suggested in [33]. To resolve this task, we applied processing of very-high-resolution imagery of various time spans for 2010–2020, covering areas with thermodenudation activities monitored in 2001–2010.

## 2. Study Area

The Yugorsky Peninsula at the southern coast of the Kara Sea (Figure 2) is a Pay-Khoy Ridge piedmont, with rolling hills, continuous permafrost zone, layers of tabular ground ice are the main features [34–36]. Two key sites are studied: the Pervaya Peschanaya key site (PP), west of the Pervaya Peschanaya River, and the Shpindler key site (Sh), west of the Hubt'Yakha River. Three TCs are identified within each site: the Western (WTC), the Central (CTC), and the Eastern (ETC).

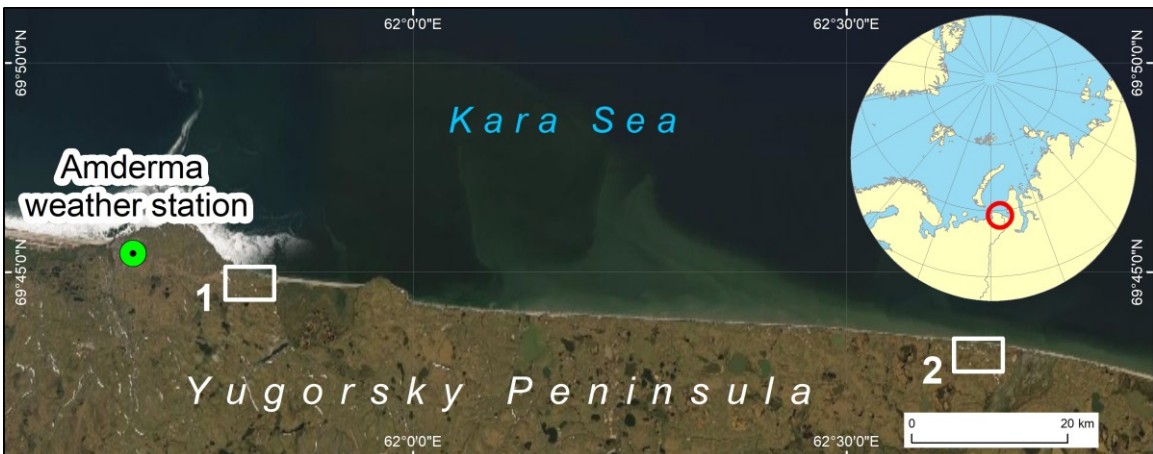

**Figure 2.** Study area: the Pervaya Peschanaya (PP) key site (1) and the Shpindler (Sh) key site (2).

### 2.1. Pervaya Peschanaya Key Site

PP occupies ancient thermodenudation depressions sloping up from the seashore relatively steeply from 20 to 45 m elevation towards the Pay-Khoy Ridge. Active TCs expose Quaternary sandy and silty deposits, enclosing one layer of tabular ground ice [27,30,35,36]. Layers of tabular ground ice exposed in the backwall of each TC are 2–6-m-thick. The edge of the backwall is 50 to 200 m away from the beach, and thus is not directly affected by the wave action (Figure 3).

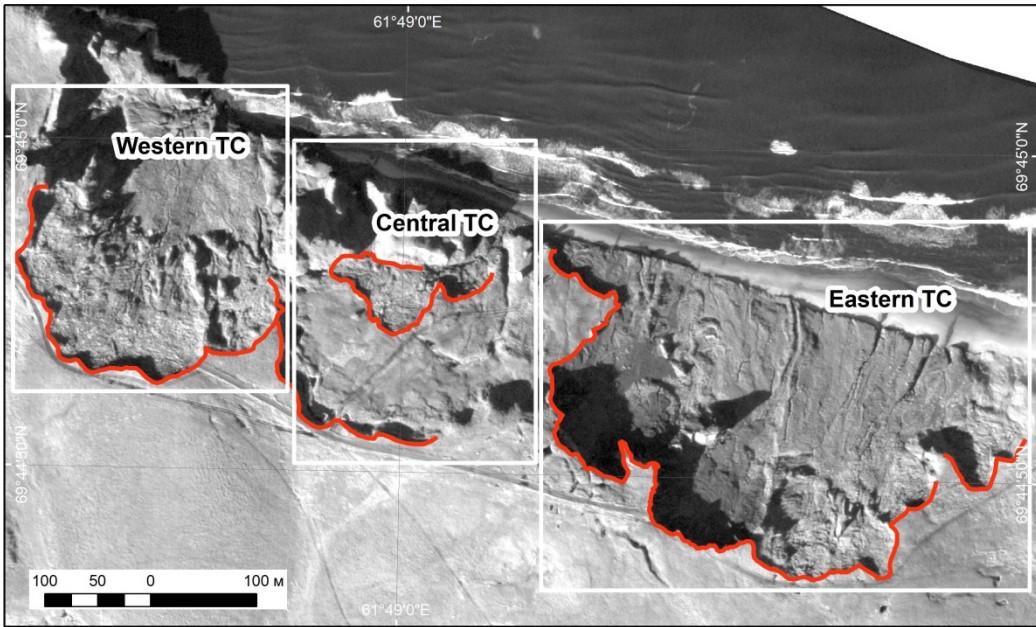

**Figure 3.** The Pervaya Peschanaya key site. The red line indicates the position of active retreat edges in 2020. WorldView-1 image dated 22 September 2020 (pan).

Before 2010, only two TCs were mapped and monitored at PP. Activation of WTC and ETC occurred in 2000–2001. From 2010–2020, three TCs are observed (Figure 3). The latest field surveys discovered that the upper limit of the tabular ground ice layer within the TC backwall actively retreating is relatively close to the surface [37,38]. Average retreat rate measured by land-based surveys along transects in 2001–2007 range at 1.6–4.2 m/year with the maximum around 5.8–6.3 m/year.

## 2.2. Shpindler Key Site

Sh is only 30 km off the PP (Figure 2) yet has a very different appearance. This site occupies a 30–45-m-high terrace composed of sandy and clay deposits enclosing two tabular ground ice layers. The upper layer is up to 12 m thick, and the lower layer is up to 5 m thick [27,34,35,39–41].

Three TCs at various states of activity were mapped here (Figure 4). Only ETC and CTC were monitored in 2001–2010, as WTC was stabilized then. After 2010, all the three TCs showed evidence of activity.

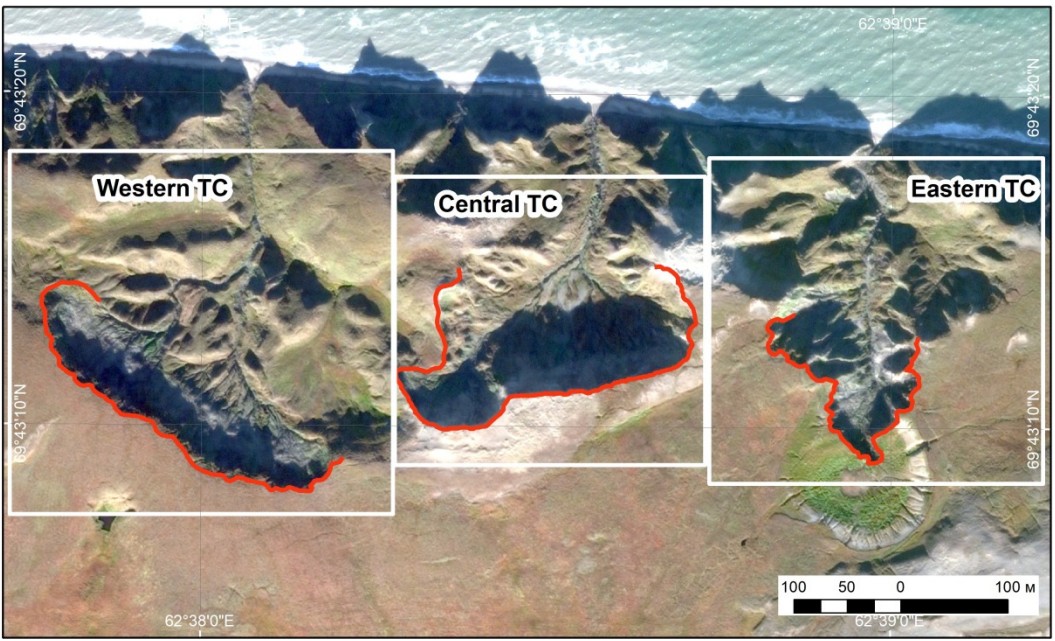

**Figure 4.** The Shpindler key site. The red line indicates the position of active retreat edges in 2020. WorldView-2 image dated 24 September 2020 (synthesis of Red+Green+Blue).

At Sh, remote-sensing data are available since 1947 (aerial survey). The average linear retreat rate for 1947–2001 determined by both remote-sensing and field-based surveys made 0.6–1.0 m/year, while, in 2001–2005, this value increased to 3.8 m/year, and in 2003–2005, it reached 7.6 m/year.

## 3. Materials and Methods

The study is based on remote-sensing monitoring of TC edge retreat and the relation of retreat rate to several climatic parameters as direct and indirect controls of thermodenudation.

## 3.1. Processing Remote-Sensing Data

We used a series of multi-temporal satellite images of the high and very high spatial resolution of 0.5 to 2.0 m/pixel, dated from 2010 to 2020 (Table 1). To improve the accuracy of orthorectification and imagery spatial alignment in areas with significant changes in surface height (areas of TC growth), we propose a methodology of DEM reconstruction.

**Table 1.** Satellite imagery used in the analysis of thermodenudation, and main retreat indicators (numbers in bold mark maximum values for each retreat indicator).

| Date | Sensor | Pan Ground Sample Distance, m | Imagery Relative Spatial Alignment (RMS), m | Monitoring Data for a Period between Acquisitions. United Data for a Group of Three TC | | | |
|---|---|---|---|---|---|---|---|
| | | | | Q 1/4, m | Median Retreat, m | Q 3/4, m | Max Retreat, m |
| **Pervaya Peschanaya key-site** | | | | | | | |
| 13 October 2010 | WorldView-1 | 0.5 | 0.3 | | | | |
| 09 October 2012 | Formosat-2 | 2.0 | 0.9 | **3.26** | **13.67** | **24.51** | **59.09** |
| 03 September 2013 | SPOT6 | 1.5 | 0.4 | 0.06 | 2.85 | 5.84 | 18.78 |
| 17 August 2015 | SPOT7 | 1.5 | 0.8 | 1.61 | 4.46 | 9.69 | 30.69 |
| 17 June 2016 | SPOT7 | 1.5 | 0.4 | 0 | 0.46 | 1.95 | 9.42 |
| 27 July 2017 | WorldView-2 | 0.5 | 0.3 | 0.625 | 2.15 | 4.65 | **60.33** |
| 22 September 2020 | WorldView-1 | 0.5 | 0.3 | 0.46 | 1.45 | 5.11 | 40.97 |
| **Shpindler key-site** | | | | | | | |
| 13 October 2010 | WorldView-1 | 0.5 | 0.3 | | | | |
| 26 June 2012 | Formosat-2 | 2.0 | 0.4 | 0 | 1.22 | 3.33 | 12.38 |
| 17 July 2013 | WorldView-1 | 0.5 | 0.3 | 0.62 | 1.67 | **5.21** | 27.23 |
| 17 August 2015 | SPOT7 | 1.5 | 0.9 | 0.085 | **1.72** | 4.78 | **40.69** |
| 17 June 2016 | SPOT7 | 1.5 | 0.57 | 0.36 | 1.22 | 3.25 | 11.7 |
| 12 October 2017 | SPOT7 | 1.5 | 0.56 | **0.405** | 1.03 | 2.02 | 25.61 |
| 24 September 2020 | WorldView-2 | 0.5 | 0.35 | 0 | 0.47 | 1.26 | 5.7 |

3.1.1. Photogrammetric Processing

The spatial alignment accuracy of multi-temporal images is important, as it determines the reliability of retrieved data on TC edge outline and retreat dynamics. High accuracy of imagery spatial alignment, as a rule, is provided by orthorectification with ground control points (GCPs) and a digital elevation model (DEM), as was carried out in [15]. However, our study area is not provided with instrumentally measured GCPs, so we used block adjustment for the spatial alignment of multi-temporal images.

The most optimal is the processing of stereopairs of satellite images for each analyzed date since it allows creating DEM consistent with satellite images, and, consequently, performing accurate orthorectification. However, due to the lack of historical images sufficient for this type of processing, we used the method of reconstructing the available DEMs in the zone of active TC or coastal retreat.

This methodological approach is relevant when images precede the available detailed DEM date. The DEM is "restored", that is, the TC areas are reconstructed in the direction from the edge to the center. Thus, we obtained a retrospective DEM with an initial (assumed) height of all analyzed lines of the TC edges. Such an approach with DEM restoration is advisable for the assessment of the coastal dynamic. This method was successfully tested in our study of the dynamics of the Kolguev Island coasts [42].

The current method is based on morphological dilation, applied to images with discrete brightness values (in our case, altitudes of the DEM matrix). The essence of the method is to assign the maximum value obtained within a structural element belonging

to a given neighborhood (local window) to the current pixel of the processed matrix. Mathematically, this can be written as follows:

$$g\ [i, j] = max\ \{W\ \{f\ [i, j]\}\}, \tag{1}$$

where $i$, $j$—the numbers of the row and column of the matrix, $g$—the reconstructed matrix, $f$— the structural element, and $W$—the local window.

The size and shape of the structural element both were selected experimentally followed by expert judgment of the results obtained. Hence, we found that the most suitable configuration for the current problem is a local window by $3 \times 3$ pixels, and the shape is as follows:

$$\begin{bmatrix} 1\ 0\ 1 \\ 0\ 1\ 0 \\ 1\ 0\ 1 \end{bmatrix}. \tag{2}$$

Convolution was applied iteratively on a manually defined area of interest, while several iterations controlled the reconstruction width.

We used ArcticDEM Strips [43] as a base for reconstruction. The ArcticDEM Strip, compiled from stereo-pair WorldView-1 of 4 September 2011, was used as a base for PP, and ArcticDEM Strips, compiled from stereo-pairs WorldView-1 and WorldView-3 of 14 June 2014 and 16 September 2015, respectively, were used as a base for Sh.

Before performing a photogrammetric task, ArcticDEM Strips processing includes: (1) masking water bodies; (2) replacing single extreme of altitude by median filtering; and (3) expert judgment, masking, and interpolation of incorrect portions of the DEM. To mask water bodies, water identification was performed, and a vector polygonal mask was created, then, the values of the DEM pixels belonging to the mask were assigned the value of the average height along the mask edge. Then, the filtering and interpolation of gaps were undertaken.

Then, reconstruction of topography for retreated parts of the coast was performed. We applied 15 iterations, which correspond to 30-m-width reconstruction (according to ArcticDEM Strips pixel size).

Photogrammetric processing includes two main stages: (1) pairwise spatial alignment of satellite images; and (2) orthorectification. Block bundle adjustment with automatic tie-point measurements and zero-order correction for RFM (WorldView data), as well as rigorous model (Formosat and SPOT data), were all used at an alignment step. Then, image verification was undertaken to prove that TC edges on all images were within the reconstructed area. Reconstructed DEMs were used at an orthorectification step and aligned multi-temporal satellite images with sub-pixel accuracy. Imagery relative spatial alignment calculated as a root mean square error for each image is shown in Table 1. Processing is performed using software SCANEX Image Processor, version 3.5.

### 3.1.2. Determining Thermocirque Dynamics (TC Edge Retreat Rates)

Coastal TC edges were digitized manually on orthorectified imagery. The TC dynamics was analyzed only in areas with active growth, where the edges were represented by a sharp line that was confidently interpreted on satellite images. Variations in spatial resolution of images and their superposition accuracy resulted in some errors drawing TC edge lines. This error for a consecutive pair of TC edges was estimated as a square root of squares' sum from Table 1. Root mean square error accuracy allowed estimating resulting spatial alignment error within 0.4 to 1.1 m for a consecutive pair of TC edges. Thus, we obtained an accurate mutual spatial alignment of multi-temporal orthorectified imagery, comparable with achieved in [15]. This allowed a detailed assessment of TC edge dynamics.

The TC edge retreat was measured according to the method applied in [15] along the transects placed every 10 m using the digital shoreline analysis system (DSAS) v5.0 add-in to Esri ArcGIS [44]. DSAS allowed the calculation of the dynamics of changes in the position of multi-temporal coastlines along a series of transects, which were automatically

built perpendicular to the baseline. This approach provides the best fit to the perpendicular of transects to both TC edge lines between which they appear. We calculated the baseline position for our TCs as Voronoi polygons between TC edge lines 2010 and 2020. Some transect lines were reoriented and reshaped as necessary to reflect the compound form of TC growth. These transects were used to measure the distances between TC edge lines using net shoreline movement (NSM) calculation. Thus, we obtained a series of TC edges retreat values for each TC.

The type of statistical distribution of the obtained values (Figures A1 and A2) does not obey the Gaussian distribution. This statement was confirmed by additional statistical tests using D'Agostino's K-squared method. The resulting *p*-value was much less than alpha (5% or 0.05) (Tables A1 and A2). So, this does not allow using the mean arithmetic when evaluating the changes in the rate of TC edge retreat with time.

Statistical measures used in this work are the median, the maximum, the first (Q 1/4), and the third (Q 3/4) quartiles following the method applied in [20]. Results are presented in box-whisker plots. Median characterizes the highest probability of TC edge linear retreat within each time interval under consideration. The interquartile range between Q1/4 and Q3/4 represents the distribution range covering the most probable retreat values, cutting off the rarest minima and maxima.

One of the objectives of the study was to compare the data obtained for 2010–2020 with previously published in [2,10,24,26–28,30–32] data for 2001–2010. In previous studies, we calculated the average rate of TC retreat as the ratio of the area between the position of the TC edges in 2001 and 2010 to the initial length of the retreated edge in 2001. For comparability with these data, we calculated the average retreat for 2010–2020 as an area enclosed between two consecutive edges, subdivided by an edge length of 2010.

### 3.2. Processing Climatic Data for 2010–2020

Data were obtained by processing the records from weather station Amderma (WMO# 23022, 69.77°N, 61.68°E). The interannual variability and main trends of air temperature and precipitation are shown on a diagram (Figure 5). One can see positive trend of air temperature and decline of atmospheric precipitation during two decades of study.

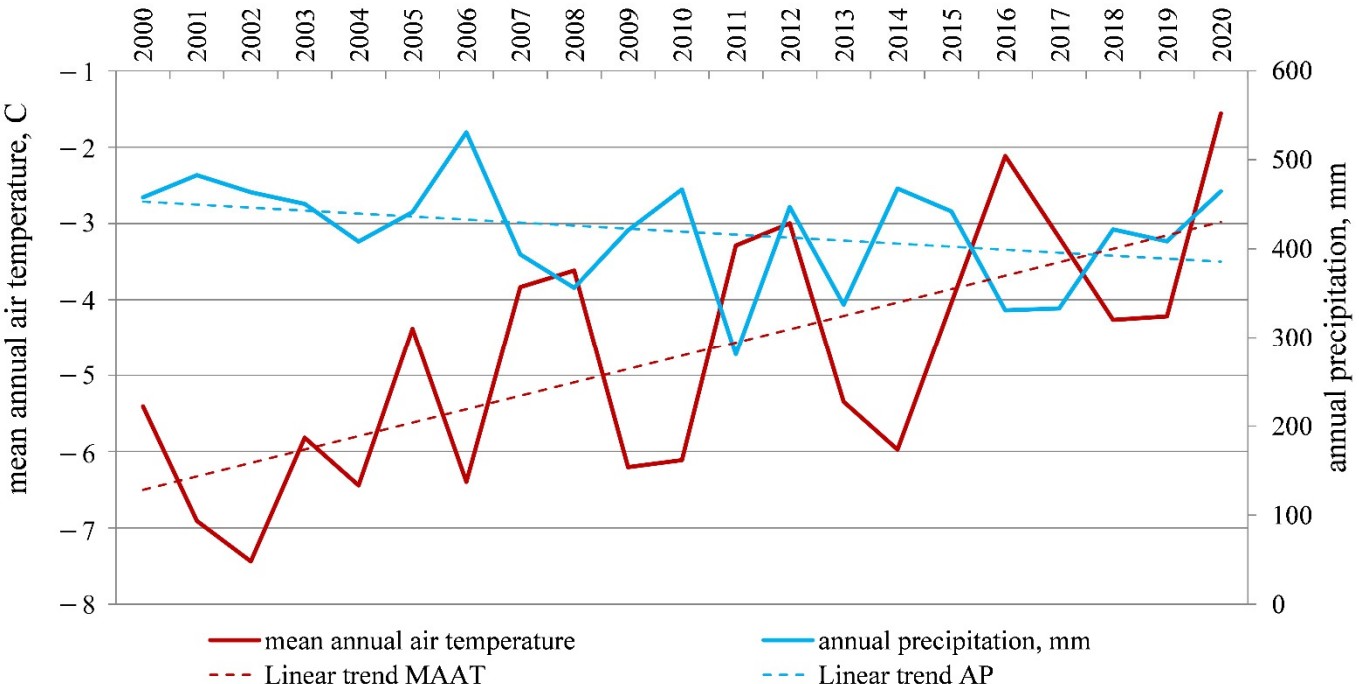

**Figure 5.** Annual climatic parameters for the 2001–2020 period, weather station Amderma.

However, we had to operate with values placed within the time interval between two adjacent acquisitions, which differ from the standard year period. So, all records were summarized for the period between acquisitions (Table 1). This time interval varied in duration for two key sites because different sets of remote-sensing data were available for each. To eliminate the difference between time intervals for two key sites, we processed climatic records, as presented in Table 2. The parameter for the entire time interval was subdivided (normalized) by the warm period duration (WPD). Parameters, both summarized for the warm period of each time interval and normalized by WPD, are thaw index, degree-days; rainfall, mm; the repeatability of northern wind (wind with a northern component, directions 0, 90 and 270, 360) with speed exceeding 10 m/s, %; and open water period duration, days. To characterize duration of open water period in this research, we used sea ice concentration data for a point with coordinates 70.02°N, 62.14°E, selected from the OSI-450 and OSI-430 satellite data archives for each time period used for coastal retreat estimates.

**Table 2.** Amderma weather station records processed for the period 2010–2020, for the warm period/normalized to warm period duration.

| Dates between Measurements | Warm Period Duration, Days | Thaw Index, degree-days | Summer Precipitation, mm | Probability North Wind Speed > 10 m/s, % | Open Water Period Duration, Days |
|---|---|---|---|---|---|
| **Pervaya Peschanaya key-site** | | | | | |
| 13.10.2010–09.10.2012 | 728 | 1937/6.5 | 472.7/**1.59** | 32.1/0.108 | 396/1.33 |
| 09.10.2012–03.09.2013 | 330 | 817/7.9 | 104.7/1.01 | 18.8/0.181 | 136/1.31 |
| 03.09.2013–17.08.2015 | 714 | 1263/5.1 | 394.2/**1.60** | 34.2/0.138 | 282/1.14 |
| 17.08.2015–17.06.2016 | 306 | 349/4.5 | 108.3/1.39 | 42.3/**0.543** | 160/**2.05** |
| 17.06.2016–27.07.2017 | 406 | 1693/**9.2** | 175.8/0.96 | 33.5/0.182 | 232/1.26 |
| 27.07.2017–22.09.2020 | **1154** | **3067**/6.3 | **732.1**/1.51 | **48.9**/0.101 | **643**/1.33 |
| **Shpindler key-site** | | | | | |
| 13.10.2010–26.06.2012 | 194 | 1109/5.7 | 267.2/1.38 | 26.5/0.136 | 291/1.5 |
| 26.06.2012–17.07.2013 | 160 | 1151/7.2 | 251.6/1.57 | 34.9/0.218 | 194/1.21 |
| 17.07.2013–17.08.2015 | 299 | 1782/6.0 | 451.2/1.51 | 31.7/0.106 | 330/1.1 |
| 17.08.2015–17.06.2016 | 78 | 349/4.5 | 108.3/1.39 | 42.3/**0.543** | 160/**2.05** |
| 17.06.2016–12.10.2017 | 259 | 2129/**8.2** | 223.4/0.86 | 39.8/0.154 | 309/1.19 |
| 12.10.2017–24.09.2020 | **410** | **2628**/6.4 | **684.5**/**1.67** | **47.8**/0.117 | **568**/1.39 |

## 4. Results

We have to operate with the available data characterizing periods of different duration between measurements, with a different set of climatic parameters, though obtained from one weather station (Table 2).

### 4.1. Overview of Retreat Pattern within the Yugorsky Coastline

The retreat rate of TC edges at two key sites changed significantly within subdivided time intervals (Figures 6–8). As seen from diagrams in Figure 8, the retreat rate is higher at PP (Figure 8, left column, bottom pane) compared to Sh (Figure 8, right column, bottom

pane). The higher range (Q1/4–Q3/4) corresponds to a higher scattering of data, and higher values of Q1/4 indicate less zero retreat transects within each TC at PP compared to Sh (Figure 8, bottom panes). So, at Sh with generally lower retreat rates, practically no stable portions of the edges are observed. In 2010–2020, medians fluctuated with the maximum in 2010–2012 at PP and 2013–2015 at Sh, minima being asynchronous as well (Figure 2).

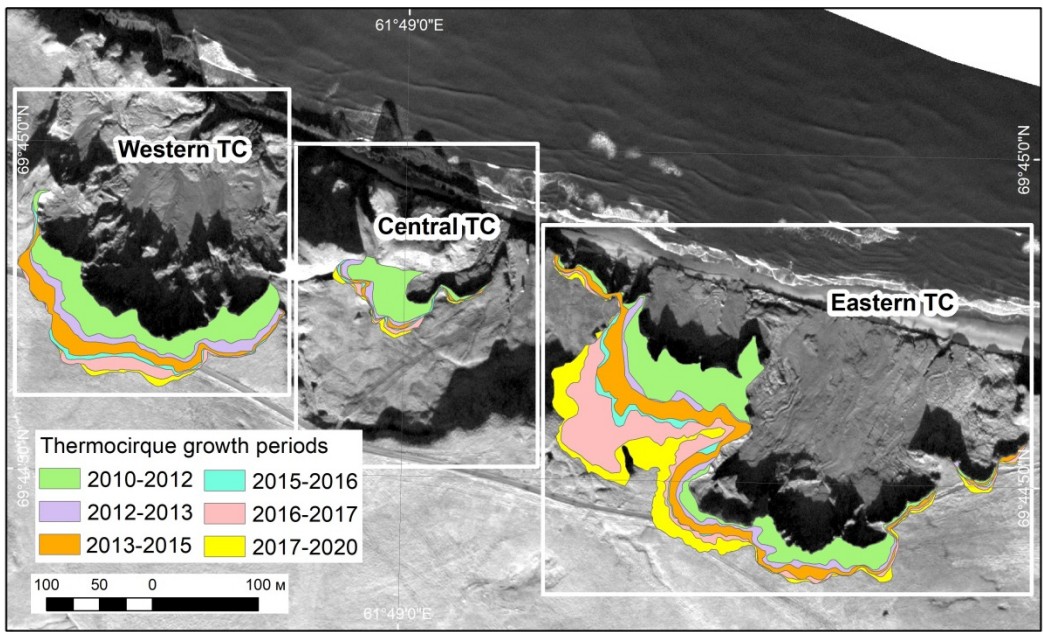

**Figure 6.** The Pervaya Peschanaya key site. The area of retreat for each time interval is shown in color.

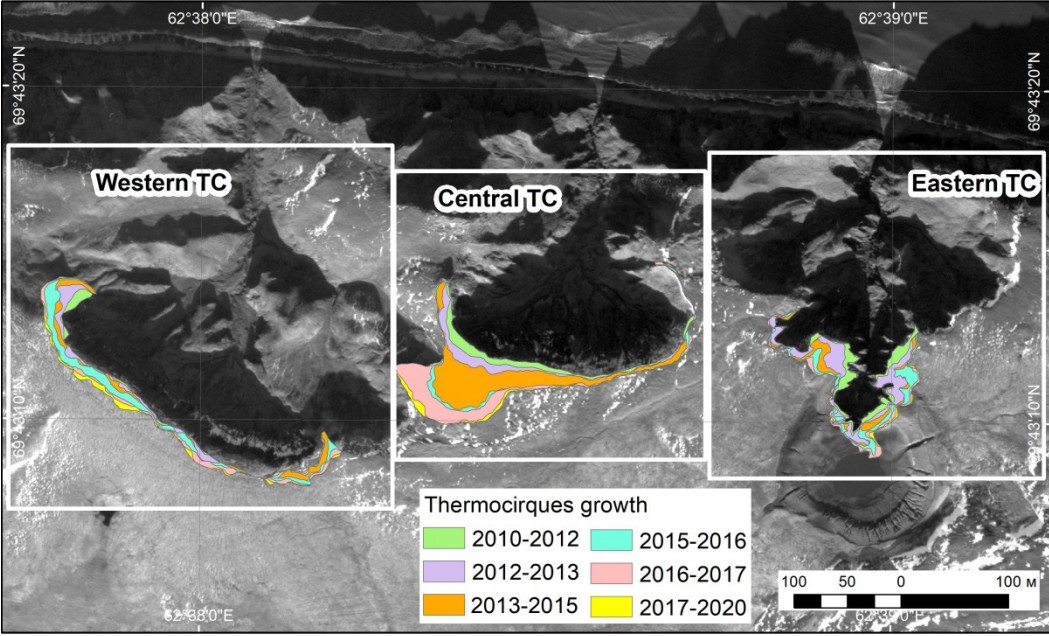

**Figure 7.** The Shpindler key site. The area of retreat for each time interval is shown in color.

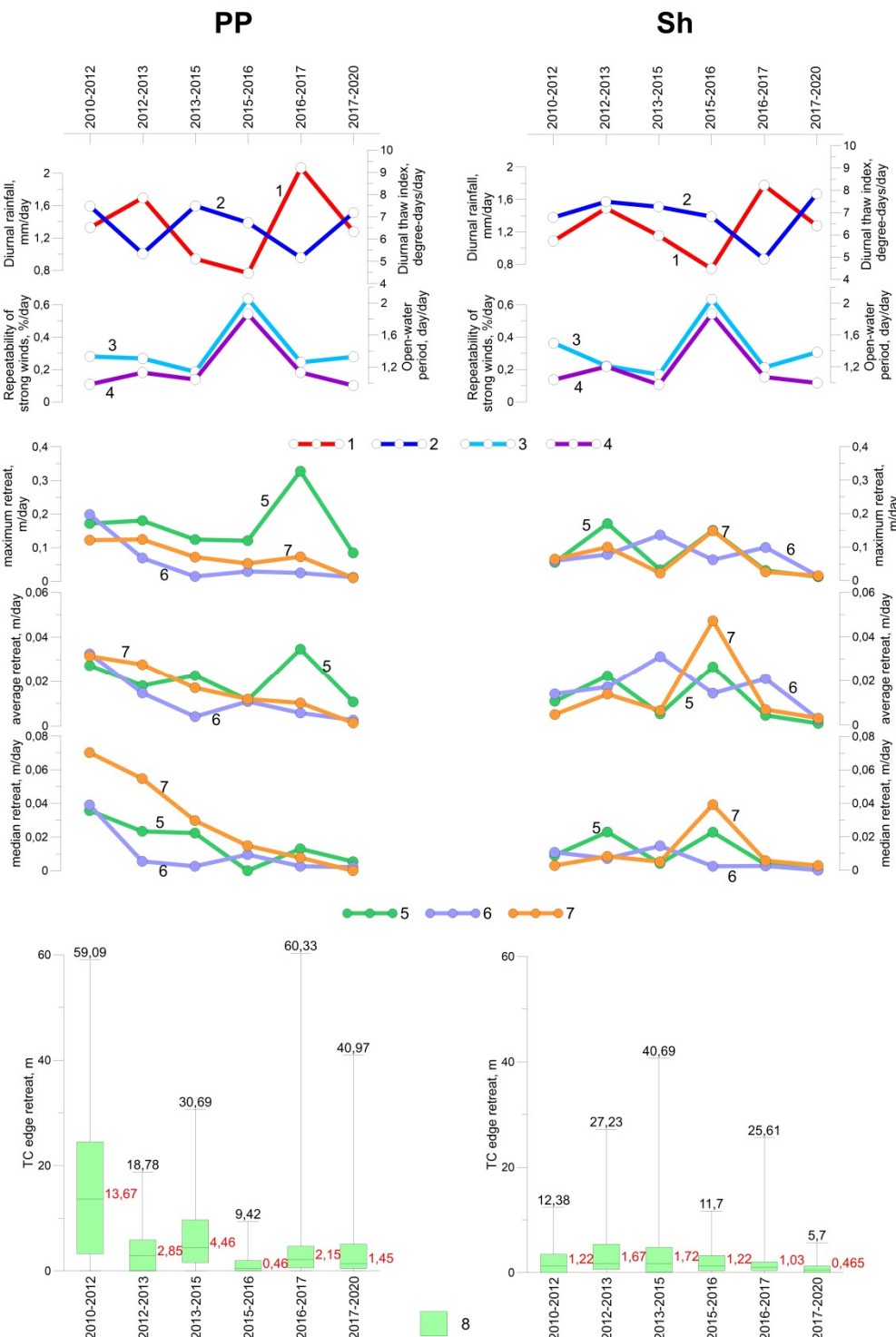

**Figure 8.** Climatic controls (upper panes) normalized by warm period duration: (1) diurnal thaw index degree-days/day; (2) diurnal rainfall mm/day; (3) normalized open-water period, day/day; (4) normalized probability of northern wind >10 m/sec, %/day; (5) normalized probability of clear sky conditions (warm period, March-September), %/day. Median, maximum, and average retreat rates, m (middle panes) for each thermocirque of both key sites separately: (6)—Eastern thermocirques, (7)—Central thermocirques, (8)—Western thermocirques. Retreat rates for TC edges: (9) box extremities (Q1/4 − Q3/4), median and maximum linear retreat, m (bottom panes) for Pervaya Peschanaya (left column), and Shpindler (right column) key sites.

For comparability with our previous observations, we compared the average linear retreat (subdividing the area of retreat by the edge length) for the entire key site and each

TC separately (Figure 8 middle panes, Tables 3 and 4). The highest rate in the median, the average, and the normalized values in all PP TCs was observed in 2010–2012. At Sh, the highest retreat rate in both median and average was observed in 2013–2015. However, when normalized by WPD, the highest value appeared within various time intervals depending on TC: WTC in 2015–2016, CTC in 2013–2015, and ETC in 2012–2013 (Figure 8, middle panes).

**Table 3.** Thermocirque edge retreat (in meters), median (Q2), maximum, and average for separate thermocirques in two key sites (numbers in bold mark maximum values for each retreat indicator).

| TC: | WTC | | | CTC | | | ETC | | |
|---|---|---|---|---|---|---|---|---|---|
| Retreat Indicator: | Q2 | Max | Average | Q2 | Max | Average | Q2 | Max | Average |
| **Pervaya Peschanaya key-site** | | | | | | | | | |
| 2010–2012 | **20.9** | **36.37** | **9.4** | **11.615** | **59.09** | **9.7** | **10.63** | **51.1** | **8.1** |
| 2012–2013 | 5.69 | 12.96 | 2.9 | 0.58 | 7.2 | 1.5 | 2.43 | 18.78 | 1.9 |
| 2013–2015 | 7.35 | 17.83 | 4.2 | 0.65 | 3.54 | 1.0 | 5.52 | 30.69 | 5.6 |
| 2015–2016 | 1.16 | 4.13 | 0.9 | 0.75 | 2.29 | 0.9 | 0 | 9.42 | 0.9 |
| 2016–2017 | 1.42 | 13.46 | 1.9 | 0.49 | 4.61 | 1.1 | 2.39 | 60.33 | 6.4 |
| 2017–2020 | 0 | 5.11 | 0.7 | 1.02 | 5.86 | 1.3 | 2.61 | 40.97 | 5.2 |
| **Shpindler key-site** | | | | | | | | | |
| 2010–2012 | 0.53 | 12.38 | 0.92 | 2.07 | 11.39 | 2.72 | 1.72 | 10.42 | 2.08 |
| 2012–2013 | 1.32 | **15.92** | 2.23 | 1.12 | 12.41 | 2.75 | **3.65** | **27.23** | **3.56** |
| 2013–2015 | 1.5 | 6.61 | 1.96 | **4.33** | **40.69** | **9.29** | 1.20 | 9.54 | 1.50 |
| 2015–2016 | **3.06** | 11.59 | **3.68** | 0.18 | 4.81 | 1.12 | 1.77 | 11.7 | 2.06 |
| 2016–2017 | 1.48 | 6.63 | 1.80 | 0.65 | 25.61 | 5.40 | 0.89 | 7.69 | 1.13 |
| 2017–2020 | 1.11 | 5.7 | 1.27 | 0 | 5.23 | 1.07 | 0 | 4.41 | 0.29 |

**Table 4.** Calculation of average retreat rate for the Pervaya Peschanaya and the Shpindler key site separately in each thermocirque for the two time intervals: 2001–2010 and 2010–2020.

| | WTC | | | | CTC | | | | ETC | | | |
|---|---|---|---|---|---|---|---|---|---|---|---|---|
| Indicator: | Retreat Area, m² | Total Edge Length in 2001 | Total Edge Length in 2010 | Active Edge Length 2010–2020 | Retreat Area, m² | Total Edge Length in 2001 | Total Edge Length in 2010 | Active Edge Length 2010–2020 | Retreat Area, m² | Total Edge Length in 2001 | Total Edge Length in 2010 | Active Edge Length 2010–2020 |
| | Pervaya Peschanaya key-site | | | | | | | | | | | |
| 2001–2010 (2010–2020) | **24,004** (13,484) | 356 | **676** | 369 | 2488 (**4004**) | 22 | 260 | 133 | 24765 (**30892**) | **638** | **1100** | **980** |
| | Shpindler key-site | | | | | | | | | | | |
| 2001–2010 (2010–2020) | 1483 (**5233**) | **432** | 441 | **417** | 8283 (**8783**) | **301** | **393** | **383** | 3007 (**5101**) | 172 | 480 | 480 |

*4.2. Pervaya Peschanaya Key Site Coastal Retreat*

For PP, the median retreat rate varied from 0.46 m during three years 2015–2016 to 13.7 m in the time interval 2010–2012. The maximum linear retreat rates of 59.1 and 60.3 m were observed in 2010–2012 and 2016–2017. The range between Q1/4 and Q3/4, in general, reduces towards 2020 with the maximum in 2010–2012. The minimum median, maximum rate, and minimum quartile range were observed in 2015–2016 (Figure 8, left column, bottom pane).

Each separate TC (Figure 6) pattern differed ETC showed a maximum retreat rate in 2016-2017, when other TCs were at their minimum. If we excludes the influence of different observation intervals, there would be no pronounced peaks, and the retreat activity would decrease almost uniformly over time (Figure 8, left column, middle pane, Table 2).

*4.3. Shpindler Key Site Coastal Retreat*

For Sh, the median retreat rate varied from 0.47 m during three years 2017–2020 and to 1.72 m in 2013–2015. In terms of one warm season, the highest median value, 1.67 m, was recorded in 2012–2013. The range between Q1/4 and Q3/4 reached its maximum in 2012–2015 and gradually reduced to 2020. The maximum linear retreat rate of 40.7 m was observed in 2013–2015. The minimum median, maximum rate, and minimum quartile range were noted in 2017–2020 (Figure 8, right column, bottom pane).

The separate TC patterns differed from both summarized deviations in Sh and PP. Sh's WTC shows two positive extremes in 2013 and 2016. CTC was active in 2014–2017 by cutting into the WTC's bottom (Figure 7). ETC shows two positive extremes in 2012–2013 and 2015–2016.

The pattern in Spindler for summarized data is complicated. A small peak in 2012–2013 was observed in the ETC, and a more pronounced peak was observed in 2015–2016 in both WTC and ETC.

## 5. Discussion

### 5.1. Comparing Retreat Rates in 2001–2010 and 2010–2020 Periods

The PP retreat area considerably reduced in 2010–2020 compared to 2001–2010 (24,004 to 13,484 m$^2$) in WTC, but considerably increased in CTC (2488 to 4004 m$^2$), and, to some extent, in ETC (from 24,765 to 30,892 m$^2$) (Table 4). Retreating edge length increased almost twice from 2001 to 2010 but then reduced again by 2020 in all thermocirques to a different extent. The retreat area at Sh increased in 2010–2020 compared to the 2001–2010 period, in WTC considerably (from 1482 to 5233 m$^2$), and in CTC only slightly (from 8283 to 8783 m$^2$), along practically the same TC edge length in both cases. The ETC retreat area increased rather noticeably (from 3007 to 5101 m$^2$) in 2010–2020 compared to the 2001–2010 period, while the edge length increased considerably in 2010 compared to 2001, remaining the same in 2010–2020.

### 5.2. Climatic Controls of Coastal Retreat

Analysis of diagrams for the area and linear retreat for both key sites, both summarized and in each TC separately (Figure 8, middle panes), were compared to climatic parameters (Figure 8, upper panes) and showed considerable differences in the rates of TC growth versus climatic parameters.

The median and maximum indicators of retreat in PP are at their peak in 2010–2012. However, this time interval is characterized by a relatively cool summer, moderate northern winds, an open water duration, and only high rainfall. The hot summer of 2012 being added to the cool summer of 2010–2011 probably did not add much to a warm peak, yet was sufficient to provide the noticeable activity of the TC edge retreat though added to less active years within the time interval (Figure 8, bottom pane, Table 2). The hot summer of 2016 is included in the period from 17 June 2016 to 27 July 2017, and affects maximum retreat rate indicator (not median and average values, though).

In Sh, peaks of retreat rate indicators are not so consistent in a pattern. Various indicators shift in time relative to each other and between various TCs. The maximum retreat rate indicator is at its peak from 26 June 2012 to 17 July 2013. The hot summer of 2012 is included here in contrast to PP. The median peak occurs from 17 July 2013 to 17 August 2015, which includes two half-summers and one whole summer. However, a medium thaw index is observed, as well as medium precipitation, and the lowest strong northern wind repeatability plus open-water duration (Figure 8, bottom pane, Table 2). Therefore, we explain the median peak only due to summarizing several years of retreat. The other close-in-value peak of the median occurs from 26 June 2012 to 17 July 2013, with one summer composed of two-year portions, including the hot summer of 2012, practically providing the highest relative median.

Detailed analysis of the retreat rates within time intervals showed the weak relation of retreat indicators to climatic parameters or their combination (Figure 9). In general, the best of low correlations belongs to the maximum retreat rate indicator versus the thaw index for PP ($R^2$=0.433). For Sh, the best fit corresponds to strong northern winds along with a high open water duration. The correlation is negative with $R^2$=0.21–0.48 for the wind repeatability versus all the retreat-rate indicators, and $R^2$=0.51 between the open water period and the median retreat rate indicator. It is hard to suggest an explanation to the effect of open water because all backwalls there are 300–500 m off the shoreline, and dropwalls are 7–12-m-high over the beach.

Consistency in the highest values of retreat parameters observed at the PP site (Table 3) compared to the inconsistent distribution of the highest retreat rates at the Sh site can be explained, in part, by the dates of activation. At PP, activation of TCs was observed in 2000–2001, while, at Sh, active TCs were already present on the aerial images of 1947. We suggest that mature TCs of Sh manifest individual rate of backwall retreat to a higher extent compared to "younger" TCs of PP.

A better correlation is obtained when studying the interrelation of climatic parameters and the number of RTS. It was noted [15] that warm summers of 2011 and 2012 are promoted over 200 RTS. However, the same publication states that there is no correlation between individual RTS retreat and warming as well as precipitation. Ref [13] determined that, during 1984–2015, the number of RTS increased 60-fold due to warm summers. Unlike Canada and Russia, in China [14], activation was observed in 2010 and 2016. This activation was mainly attributed to anomalously high summer temperature and abundant precipitation. Intensive rainfall is a reason for significant activation [5]. The increase in TC number in Central Yamal was related to the warm summer of 2012, but retreat rates did not increase in the even warmer summer of 2016 [12,45], which is the same as in the study area on the Yugorsky peninsula.

Other climatic controls of the coastal retreat are also a matter of analysis in several papers. The highest coastal retreat rates are measured on the north-west-facing shorelines, which correspond to strong waves' main direction [46]. Relation of retreat rate to scarp height was found in [47]. They also established that slope orientation does not affect the retreat rate of RTS. In the mountainous regions of the Canadian Arctic, the slope aspect plays an essential role in RTS development. South- and west-facing slopes show higher retreat rates [33]. Ice thickness was mentioned as an essential factor of RTS distribution, while RTS activity and initiation depend on a slope angle [7]. The impact of ice thickness for PP was proved at the previous monitoring stage [30].

All weaknesses in the relationship between retreat rates and climatic parameters should be attributed to other factors, such as relief, environment, ice content, and distribution in the section. Environmental controls of retreat in the study area were suggested [32] and relatively reliably show the retreating edges of TCs in 2010–2020 expansion into less resistant environmental units.

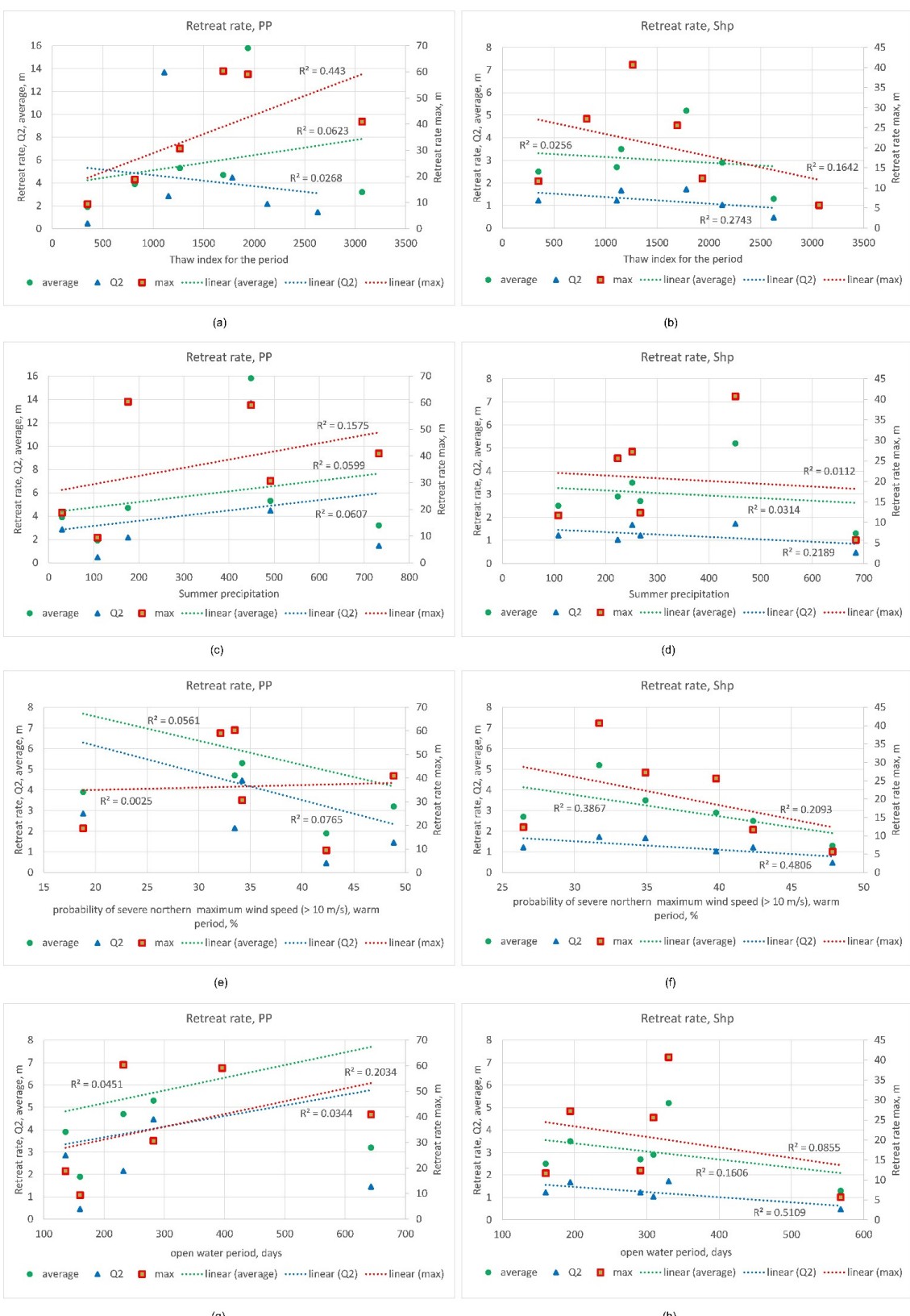

**Figure 9.** Relation of retreat rate indicators (median, maximum, and average) summarized for a key site, to climatic controls presented in Table 2. The left pane for the Pervaya Peschanaya key site, the right pane for the Shpindler key site: (**a**,**b**) average, median and maximum thaw index, degree days; (**c**,**d**) summer precipitation, mm; (**e**,**f**) probability of summer wind speed >10 m/s, %; (**g**,**h**) open water period, days.

*5.3. Comparing Retreat Rates in the Yugorsky Peninsula and Other Regions of the World*

The long-term retreat rate calculated from the analysis of remote-sensing data at the Yugorsky coast before 2010 [24] was close to that obtained for various arctic areas, approximately 1 m/year for the period 1948–2001. Somewhat similar areas of the Canadian Arctic show a retreat rate of 1.03 m/year in 1970–2000 [23]. Though records averaged for an extended period (53 years for the Yugorsky Peninsula and 30 years for Canada) show relatively low values, annual data display much higher rates.

5.3.1. Regions Farther North from the Yugorsky Peninsula

In the Eureka Sound Lowlands area, the Ellesmere and Axel Heiberg Islands, the average individual TC retreat rates in 2011–2018 comprised 0–26.7 with a maximum of 79 m/year [15]. The maximum rates were noted in 2011–2012 due to high summer temperature. At Yugorsky, in the same period of 2010–2012, an intensive growth of TCs and the first peak of maximum rates (59.1 m for the period) were recorded at the PP. After the peak of 2011–2012, the growth rates in most RTS decreased by 2018 [15]; in the Yugorsky Peninsula, the trend of decreasing thermodenudation rates is similarly observed.

5.3.2. Regions at the Same Latitude or South of the Yugorsky Peninsula

On Herschel Island in 2001–2004, the long-term average growth rates of TCs (RTS) were 9.6 m/year, the maximum being 18 m/year [23]. From 2004–2006, the maximum growth rates decreased to 4–10.5 m/year [48]. The maximum rates are lower, and the average perennial ones are higher than on the Yugorsky Peninsula.

Individual rates for 18 TCs of the northern Mackenzie valley in 2007–2009 varied from 0 to 15.1 m/year [47]. In the Richardson Mountains and Peel Plateau for the period 1990–2010, the average annual growth rates of individual TCs was within the range of 5 to 60 m/year. Averaged for a 20-year period, the rates were 7.2–26.7 m/year [33]. The retreat rate reached 38 m/year in the Noatak Valley of northern Alaska in 2010–2016 [49]. The annual rates of individual TCs, the same as in Yugorsky, show a significant scattering, and the values averaged over a long-time interval exceed the average values for the Yugorsky key sites (3.4 m/year for 2010–2020).

# 6. Conclusions

We applied a method for processing multi-temporal satellite images that ensure the accurate alignment of analyzed images. Thus, we calculated the dynamics of TC edges with sub-meter accuracy even without support by land-based surveys.

Measuring the TC edge retreat for several time intervals from 2010–2020 showed an oscillatory character of TC development. In contradiction with our previous studies and some published observations, retreat rates of TC edges do not depend on any climatic drivers at least for the study period. Maximum retreat shows stronger dependence on the thaw index than average values. High medians coinciding with the peak of maximums for a given time interval reinforce the superposition of favorable climatic conditions on the favorable environmental features of the area. The maximum retreat rates, not accompanied by high medians within a time interval, suggest the superposition of high summer warmth and relatively unfavorable environmental conditions along the retreating edge with only local exceptions producing complicated configuration of the retreating TC edge characteristic of the Yugorsky Peninsula.

Different TCs in two key sites develop at their rate, some activate in much warmer but are drier in 2010–2020 compared to 2001–2010; however, in general, they stabilized within a two-decade time interval. Stabilization is most likely caused by increased accumulation of thawed sediments insulating ice exposures, but may also respond to changes in ice layer thickness reduction at different portions of the coastal plain.

**Author Contributions:** M.L. wrote the manuscript with contributions from A.K. and M.Z., A.K. gathered and prepared all data with contributions from M.Z. and A.S., and made all figures. M.Z. developed

the methodology with input from A.S. and A.K., M.L., A.K., M.Z., and A.S. contributed to the discussion of results. Y.Z. collected and processed all climatic data and contributed to the discussion. Funding acquisition, M.Z. All authors have read and agreed to the published version of the manuscript.

**Funding:** This research was mainly funded by the Russian Foundation of Basic Research grant № 18-05-60221 and partially funded by the Russian Foundation of Basic Research № 18-05-60222, State Assignments 121051100164-0, 121051400061-9 and 121041600042-7. The APC was funded by the Russian Foundation of Basic Research grant № 18-05-60221.

**Data Availability Statement:** Publicly available climatic dataset was analyzed in this study. This data can be found here: http://meteo.ru/, accessed on 20 June 2020, weather station #23022, Amderma. Remote-sensing data was obtained from Airbus Defence, Maxar and Scanex due to license agreements, and are available at https://www.airbus.com/space.html, accessed on 20 June 2020; https://www.maxar.com/, accessed on 20 June 2020; and https://www.scanex.ru/en/, accessed on 20 June 2020, with the permission of Airbus Defence, Maxar, and Scanex, respectively.

**Acknowledgments:** Centre of collective usage «Geoportal», Lomonosov Moscow State University provided access to remote sensing data.

**Conflicts of Interest:** The authors declare no conflict of interest. The funders had no role in the design of the study; in the collection, analyses, or interpretation of data; in the writing of the manuscript, or in the decision to publish the results.

## Abbreviations

A list of non-standard abbreviations:

| | |
|---|---|
| RTS | Retrogressive thaw slump |
| TC | Thermocirque |
| PP | Pervaya Peschanaya key site |
| Sh | Shpindler key site |
| DEM | Digital elevation model |
| WPD | Warm period duration |
| WTC | Western Thermocirque |
| CTC | Central Thermocirque |
| ETC | Eastern Thermocirque |

## Appendix A

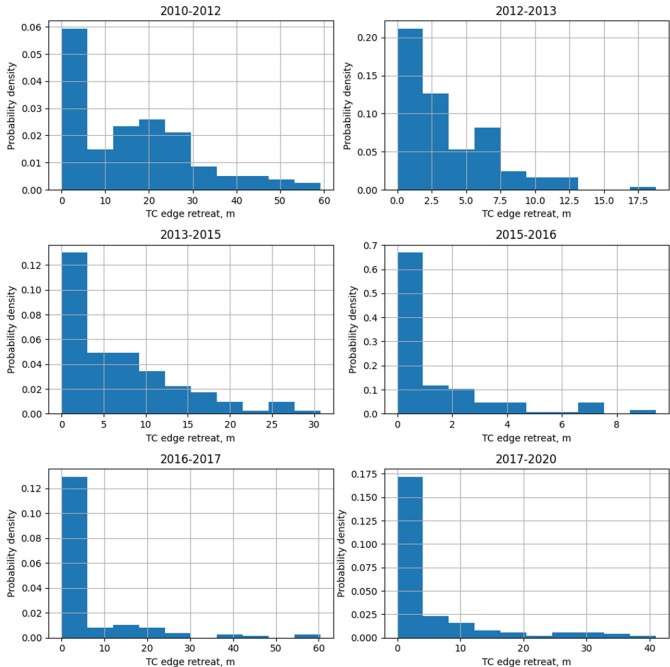

**Figure A1.** Distribution histograms for a series of transect lengths in Pervaya Peschanaya key site.

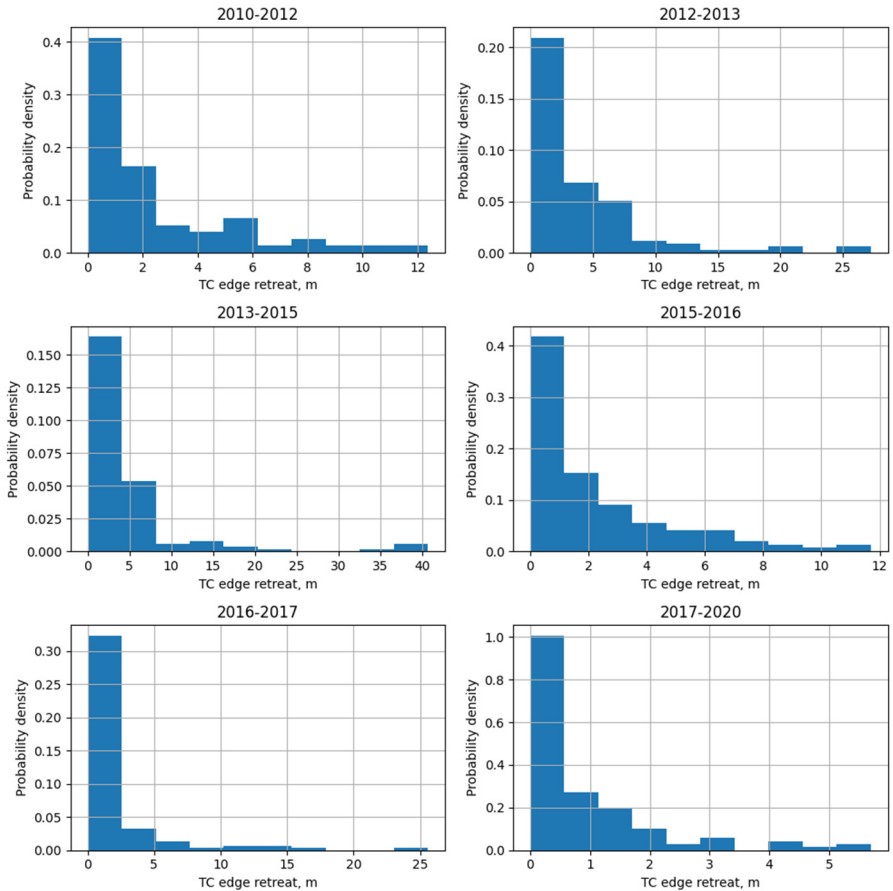

**Figure A2.** Distribution histograms for a series of transect lengths in the Shpindler key site.

**Table A1.** Statistical test for normality of distribution for a series of transect lengths in Pervaya Peschanaya (PP) key site.

|  | **2010–2012** | **2012–2013** | **2013–2015** | **2015–2016** | **2016–2017** | **2017–2020** |
|---|---|---|---|---|---|---|
| count | 137 | 131 | 133 | 135 | 127 | 125 |
| mean | 16.084891 | 3.534504 | 7.006541 | 1.364963 | 6.076142 | 5.18736 |
| std | 14.033858 | 3.521240 | 6.807610 | 2.068945 | 10.709844 | 8.551684 |
| min | 0.0 | 0.0 | 0.0 | 0.0 | 0.0 | 0.0 |
| 5% | 0.342 | 0.0 | 0.22 | 0.0 | 0.0 | 0.0 |
| 25% | 3.26 | 0.06 | 1.61 | 0.0 | 0.625 | 0.46 |
| 50% | 13.67 | 2.85 | 4.46 | 0.46 | 2.15 | 1.45 |
| 75% | 24.51 | 5.84 | 9.69 | 1.945 | 4.65 | 5.11 |
| 95% | 44.44 | 10.015 | 20.426 | 6.754 | 27.11 | 27.88 |
| max | 59.09 | 18.78 | 30.69 | 9.42 | 60.33 | 40.97 |
| Normal | Test | alpha = 0.05 | | | | |
| 2010–2012 | alpha = 0.05 | result = 0.0004 | | | | |
| 2012–2013 | alpha = 0.05 | result = 0.0 | | | | |
| 2013–2015 | alpha = 0.05 | result = 0.0 | | | | |
| 2015–2016 | alpha = 0.05 | result = 0.0 | | | | |
| 2016–2017 | alpha = 0.05 | result = 0.0 | | | | |
| 2017–2020 | alpha = 0.05 | result = 0.0 | | | | |

**Table A2.** Statistical test for normality of distribution for a series of transect lengths in Shpindler key site.

|  | 2010–2012 | 2012–2013 | 2013–2015 | 2015–2016 | 2016–2017 | 2017–2020 |
|---|---|---|---|---|---|---|
| count | 123 | 123 | 123 | 123 | 122 | 122 |
| mean | 2.259268 | 3.720325 | 4.266423 | 2.255366 | 2.078852 | 0.852705 |
| std | 2.85189 | 4.960016 | 7.525415 | 2.608869 | 3.592606 | 1.202779 |
| min | 0.0 | 0.0 | 0.0 | 0.0 | 0.0 | 0.0 |
| 5% | 0.0 | 0.0 | 0.0 | 0.0 | 0.0 | 0.0 |
| 25% | 0.0 | 0.62 | 0.085 | 0.36 | 0.405 | 0.0 |
| 50% | 1.22 | 1.67 | 1.72 | 1.22 | 1.03 | 0.465 |
| 75% | 3.33 | 5.205 | 4.78 | 3.25 | 2.0175 | 1.255 |
| 95% | 8.312 | 12.815 | 16.273001 | 7.652000 | 7.637 | 3.357 |
| max | 12.38 | 27.23 | 40.69 | 11.7 | 25.61 | 5.70 |

| Normal | Test | alpha = 0.05 |
|---|---|---|
| 2010–2012 | alpha = 0.05 | result = 0.0 |
| 2012–2013 | alpha = 0.05 | result = 0.0 |
| 2013–2015 | alpha = 0.05 | result = 0.0 |
| 2015–2016 | alpha = 0.05 | result = 0.0 |
| 2016–2017 | alpha = 0.05 | result = 0.0 |
| 2017–2020 | alpha = 0.05 | result = 0.0 |

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
