# Peer review of "Coastal Retreat Due to Thermodenudation on the Yugorsky Peninsula, Russia during the Last Decade, Update since 2001–2010"

_remotesensing, doi:10.3390/rs13204042_

Round 1

Reviewer 1 Report

The manuscript “Coastal Retreat Due To Thermodenudation On Yugorsky Pen-2 insula, Russia During The Last Decade, Update Since 2001-2010” suggest a new methodology to calculate the retreat rate of the thermocirque edge using various statistical approaches.

The title is appropriate for the content and the article is well constructed, written, clear and presented. The Introduction provides a good background of the topic that quickly gives the reader an overview and clearly stated objective and purpose of the performed research.

The methodology is practices on 2 different sites examples in Yugorsky peninsula coast. Methods are appropriate, thus the method of determining the retreat rate of the thermocirque edge through satellite images is quite simple, but not wrong. In my opinion, is very appropriate and interesting the analysis of “Climatic controls of coastal retreat”. The methodology used by the authors has a strong regional application and is also applicable in many other similar landscapes with particular reference to hard-to-access areas of the world.

I recommend this manuscript to be accepted and published after the minor revision concerning comments below:

  1. Check in figure 9 the words written in Cyrillic alphabet and replace them in English language; also check for text overlaps (e.g. fig. 9 - g).
  2. In Appendix A: For easy reading, I would suggest including the units for x-axis and y-axis.

Author Response

Remotesensing-1392725 response to reviewers’ comments

We highly appreciate reviewers’ comments and inserted corrections needed. Corrections made in the text in “track changes”. Corrected figures are replaced in the text and included separately to the RS1392725_Figures corrected.zip file.

Reviewer 1

  1. Check in figure 9 the words written in Cyrillic alphabet and replace them in English language;

Resp. Thank you for the comment. Corrections to Fig.9 made.

  1. Сheck for text overlaps (e.g. fig. 9 - g).

Resp. Thank you for the comment. Corrections made.

  1. In Appendix A: For easy reading, I would suggest including the units for x-axis and y-axis.

Resp. Thank you for the comment. Units added.

Reviewer 2 Report

Dear authors,

Thank you very much for your manuscript.

In my opinion, the authors should consider the following points when revision:

  • In the part ‘Determining thermocirque dynamics’ how solve the problem with the angle between chosen transect and coastline? I mean in order the transect and the coastline have a very acute angle (sub-parallel to transect) as the results one can greatly overestimated values of coastal dynamics. Have the values of TC edge retreat rates been recalculated taking angle into account?
  • Line 225, please check the format.
  • The thermodenudation plays a critical role in the Arctic coastal erosion. Slope exposure is one of the main factor of thermodenudation. Did you taken in account slope exposure for studied sites? In my opinion, it would be interesting to see how the temperature affects to thermocirque dynamics with account of slope orientation (slope exposure).
  • Line 307, probably it should be ‘2000 – 2010’ instead ‘200-2010’
  • Bad quality of Figure 9.
  • Why only the Canadian Arctic is taken in consideration? There are many studies on the same topic for Kara sea region. Is this site unique for the Kara Sea coastline?
  • References list did not draw up according to the template.
  • Please add axis’s captions on diagrams of figure A1 and figure A2.

Best regards

Author Response

Remotesensing-1392725 response to reviewers’ comments

We highly appreciate reviewers’ comments and inserted corrections needed. Corrections made in the text in “track changes”. Corrected figures are replaced in the text and included separately to the RS1392725_Figures corrected.zip file.

Reviewer 2

1. In the part ‘Determining thermocirque dynamics’ how solve the problem with the angle between chosen transect and coastline? I mean in order the transect and the coastline have a very acute angle (sub-parallel to transect) as the results one can greatly overestimated values of coastal dynamics. Have the values of TC edge retreat rates been recalculated taking angle into account?

Resp. Thank you for the comment. In the text, we added more to the explanation of  the algorithm of choosing the best position of the transects. A series of transects were automatically arranged perpendicular to the baseline, which is a line closest to the curvature of both TC edges between which the transects were built. We calculated the baseline position for our TCs as Voronoi polygons between TC edge lines 2010 and 2020. Thus, the transects appear to be the closest to perpendicular to both TC edges for the given period. 

  1. Line 225, please check the format

Resp. Corrections made.

  1. Did you take in account slope exposure for studied sites? In my opinion, it would be interesting to see how the temperature affects to thermocirque dynamics with account of slope orientation (slope exposure).

Resp. Thank you for the comment. In our key sites coast is oriented similarly for all thermocirques to the north, so in this paper we do not consider this forcing factor. However, our Yamal studies as well as published Canadian results prove that there is no direct relation between slope aspect and RTS activity.

  1. Line 307, probably it should be ‘2000 – 2010’ instead ‘200-2010’

Resp. Thank you for the comment. Corrections made.

  1. Bad quality of Figure 9.

Resp. Thank you for the comment. Figure 9 is re-worked, Cyrillic replaced by English, resolution increased to 600dpi.

  1. Why only the Canadian Arctic is taken in consideration? There are many studies on the same topic for Kara sea region. Is this site unique for the Kara Sea coastline?

Resp. Thank you for the comment. Almost all the studies of the Kara sea are presented by our group or our coauthors. Few studies of the Yamal coast of the Kara sea and east of our key sites at Yugorsky coast are devoted to massive ground ice properties, palaeo-geography of the geological sections, to coastal thermoerosion (thermoabrasion in Russian terminology), but not to RST (thermodenudation or thermocirques in Russian terminology).

  1. References list did not draw up according to the template.

Resp. Thank you for the comment. Corrections made.

  1. Please add axis’s captions on diagrams of figure A1 and figure A2.

Resp. Thank you for the comment. Corrections made, axis captions added.